# B-cell activating factor (BAFF) and its receptors' expression in pediatric nephrotic syndrome is associated with worse prognosis

**Jessica Forero-Delgadillo**[1], **Vanessa Ochoa**[1], **Jaime M. Restrepo**[1,2], **Laura Torres-Canchala**[3], **Ivana Nieto-Aristizábal**[3], **Ingrid Ruiz-Ordoñez**[3], **Aura Sánchez**[4], **María Claudia Barrera**[5], **Carlos Andrés Jimenez**[4], **Gabriel J. Tobón**[5,6]*

**1** Facultad de Ciencias de la Salud, Universidad Icesi, Cali, Colombia, **2** Servicio de Nefrología Pediátrica, Fundación Valle del Lili, Cali, Colombia, **3** Centro de Investigaciones Clínicas, Fundación Valle del Lili, Cali, Colombia, **4** Servicio de Patología, Fundación Valle del Lili, Cali, Colombia, **5** Universidad Icesi, CIRAT: Centro de Investigación en Reumatología, Autoinmunidad y Medicina Traslacional, Cali, Colombia, **6** Laboratorio de Inmunología, Fundación Valle del Lili, Cali, Colombia

* gtobon1@yahoo.com

**Data Availability Statement:** The dataset generated and/or analyzed during the current study are not publicly available in order to protect patient

## Abstract

### Aim

Immune pathogenesis of nephrotic syndrome (NS) is not completely understood. We aimed to evaluate the expression of B-cell activating factor (BAFF) and its receptors in renal samples from pediatric NS patients and its relationship with renal function survival.

### Materials and methods

We conducted an ambispective study on 33 patients with pediatric NS. Immunohistochemistry for BAFF, TACI, BCMA and BR3 was performed. Markers were evaluated on podocytes and interstitial inflammatory infiltrates (III). We performed Kaplan-Meier curves to describe renal function survival according to markers' expression.

### Results

Thirty-three NS patients were included. Minimal change disease was seen in 21 (63.6%) patients, and focal segmental glomerulosclerosis in 12 (36.4%). BAFF was found in podocytes (18.2% of samples) and III (36.4% of samples), BAFF-R in one sample, TACI in 4 (podocytes and III), and BCMA in 5 samples of podocytes and 7 of III. BAFF on podocytes and III was associated with worst renal function at follow-up; those patients had 25% probability of having GFR >90 mL/min/1.73m$^2$, versus 84.9% when absent **(p = 0.0067)**. Patients with BAFF in III had 42.9% probability of having GFR>90 mL/min/1.73 m$^2$, versus 94.1% when absent **(p = 0.0063).**

### Conclusion

BAFF expression in renal biopsies could be a prognostic factor for renal function.

information in accordance with Declaration of Helsinki but are available from the corresponding author on reasonable request. In that case, please contact with publications area of Fundación Valle del Lili clinical research center, through the email publicaciones@fvl.org.co to process your request.

**Funding:** Author: R, JM. Grant number: CA0513114 Funder: Universidad Icesi. URL: https://www.icesi.edu.co/es/ Role: The funders had no role in study design, data collection and analysis, decision to publish, or preparation of the manuscript.

**Competing interests:** The authors have declared that no competing interests exist.

**Abbreviations:** Nephrotic syndrome, NS; Minimal change disease, MCD; Focal segmental glomerulosclerosis, FSGS; Chronic kidney disease, CKD; Tumor necrosis factor family, BAFF; Transmembrane activator and CAML interactor, TACI; Protein maturation of B cells, BCMA; BAFF-receptor or BR-3, BAFF-R; Systemic lupus erythematosus, SLE; Glomerular filtration rate, GFR; Interstitial inflammatory infiltrates, III.

## Introduction

Nephrotic syndrome (NS) is a renal disease characterized by increased permeability of the glomerular filtration barrier and is clinically manifested by proteinuria, hypoalbuminemia, edema and hyperlipidemia. In addition, a hypercoagulable state may be present due to hypoalbuminemia [1]. NS can affect children of any age and is the most common in school-aged children and adolescents, with an incidence of 1.1 to 7 per 100,000 children and a worldwide prevalence of 16 cases per 100,000 [2, 3].

The most common causes of NS in pediatric patients are minimal change disease (MCD) in 80% of cases (associated with good prognosis) and focal segmental glomerulosclerosis (FSGS) in 10–15% of cases. The latter is associated with a worse prognosis and risk of progression to chronic kidney disease (CKD).

Several classifications, in addition to histological class, have been made, including idiopathic NS *vs.* secondary; genetic *vs.* acquired; and steroid-sensitive *vs.* steroid-resistant patients [4]. Steroids are the gold-standard therapy in patients with NS. Approximately 80–90% of pediatric patients with NS respond to therapy and maintain normal renal function [5]. However, up to 70% of patients experience a relapsing course or progress to a cortico-resistant or corticosteroid-dependent disease [5]. Failure to respond to corticosteroids is predictive of non-MCD histology, and among these patients, up to 50% will progress to CKD [5]. In patients with failure to respond to corticosteroids or in a corticosteroid-dependent course, a renal biopsy is recommended to establish the renal lesion, define new therapeutic options and predict prognosis [6, 7]. Additional immunosuppressive treatment includes calcineurin inhibitors (cyclosporine and tacrolimus), mofetil mycophenolate and rituximab.

The cause of idiopathic NS remains unknown. The evidence demonstrates that podocytes are the target cells of an underlying immune mechanism that leads to podocyte dysfunction. According to this concept, several authors suggest that MCD and FSGS represent a spectrum of the same immunological disease [8]. The proposed immune mechanisms involved both T and B lymphocytes. For instance, T cells may produce cytokines that affect glomeruli and cause increased permeability, as well as the production of circulating factors that alter podocyte structure and functions, resulting in proteinuria [9]. On the other hand, B-cell involvement is also suggested based on remission after treatment with B-cell-depleting therapies, such as rituximab [10].

However, the role of B-cell survival factors and cytokines in the pathogenesis of NS is not well known. The B-cell activating factor belonging to the tumor necrosis factor family (BAFF) is an essential cytokine for the maturation and survival of B lymphocytes, as well as for peripheral development and activation of lymphoid organs. This cytokine exerts its function through interactions with three receptors: Transmembrane activator and CAML interactor (TACI), protein maturation of B cells (BCMA) and BAFF-receptor or BR-3 (BAFF-R), whose expression is restricted to B and T lymphocytes [11].

BAFF and its receptors have been described as mediators of the response in immune-mediated diseases, such as systemic lupus erythematosus (SLE) [12] and pediatric SLE [13]. In addition, we have previously demonstrated that the expression patterns of BAFF and its receptors differ according to lupus nephritis class, where it is more involved in Class IV with high activity indexes [14].

As the role of BAFF and its receptors in pediatric patients with NS has not been evaluated, we aimed to investigate the expression of BAFF and its three receptors in kidney biopsies from patients with NS and to correlate the expression with clinical outcomes.

## Materials and methods

### Patients

This ambispective study (2011 to 2019) was conducted using all available kidney samples from pediatric patients with NS (0 to 18 years) followed by the pediatric nephrology service at the Fundación Valle del Lili (a tertiary care university clinic with a pediatric nephrology service that attends 3000 patients/year and a catchment area of approximately 4.6 million people in the southwest area of Colombia). No other diagnosis than idiopathic NS are included. This study was conducted following the Declaration of Helsinki statements and was approved by the institutional IRB (#1283).

Demographic, clinical, laboratory and immune variables were collected at the diagnosis of NS and at the kidney biopsy procedure. Renal biopsy was indicated by clinicians when patients failed to respond to corticosteroid therapy or in the case of relapsing and/or resistant disease as follows: **corticosteroid dependent** (relapse during taper or within 2 weeks of discontinuation of corticosteroid therapy); **relapse disease** (relapsing of severe proteinuria $\geq 40$ mg/m$^2$/h or dipstick reading of $\geq 2 +$ for 3 of 5 consecutive days; **frequent relapse** ($\geq 4$ relapses in any 12-month period); and NS **corticosteroid resistant** (inability to induce remission within 4 weeks of daily corticosteroid therapy) [1].

The main outcome variable was the glomerular filtration rate (GFR) at the last follow-up. Renal failure was defined as a GFR less than 90 mL/min/1.73 m$^2$ according to the KDIGO. High blood pressure (defined as systolic and/or diastolic blood pressure above the 95$^{th}$ percentile for age [6]), proteinuria in the nephrotic range (urine protein:creatinine $> 2$), the need for renal replacement therapy, renal transplantation and death were considered secondary endpoints.

### Immunohistochemistry

Immunohistochemistry was performed on 33 biopsy samples from pediatric patients with NS. The immunoperoxidase technique was used. Briefly, 3-mm sections of kidney tissue were collected on slides, followed by deparaffination at 60°C for 30 minutes and rehydration of samples with xylol, N-propanol, 95% ethanol, and water. The detection system EnVision FLEX (Dako) was used, according to the following steps: (1) blocking of endogenous peroxidase in PT Link with Target Retrieval Solution High pH (50x) (for BCMA, TACI and BAFF) or with Target Retrieval Solution Low pH (50x) (for BAFF-R), at 94°C for 20 minutes; (2) primary antibody incubation for 40 minutes in a cold chamber, with the following dilutions: 1:250 for BCMA (rabbit polyclonal, GTX16222), 1:100 for TACI (clone C-9, Santa Cruz); 1:125 for BAFF (Clone Buffy-2, GTX16081); and 1:50 for BAFF-R (clone 11C1, Santa Cruz). Following incubation with an EnVision FLEX/HRP secondary antibody at room temperature for 20 minutes, DAB was probed for 5 minutes, and samples were counterstained with Harris hematoxylin. Tonsil tissue was used as a positive control. Between each step, after endogenous peroxidase quenching, washes were performed with FLEX Wash Buffer.

### Statistical analysis

Slides were read by one expert nephropathologist blinded to clinical outcome, and samples were scored according to the percentage of cells with positive expression on podocytes and interstitial inflammatory infiltrates (III) (mononuclear cells, B or T-cells), as follows: 1) 1–25% of cells stained; 2) 26–50% of cells stained; and 3) more than 50% of cells stained. Dichotomous variables are reported as percentages. Continuous variables are presented as medians and interquartile ranges or means and standard deviations according to the normality of their

distribution. The probability of renal function equal to or greater than a GFR of 90 mL/min/ 1.73 $m^2$ at 60 months was estimated using the Kaplan-Meier method according to BAFF and receptor expression either in the podocytes or III in kidney biopsy. A *p* value 0.05 was considered significant. The analyses were carried out with the Stata® 14.0 (StataCorp, 2014, College Station TX, USA) statistical package.

## Results

### Patients

From 2011 to 2019, 75 pediatric patients with NS were diagnosed and followed up in our center.

Of them, 33 had a kidney sample, representing the final number of included patients in the current analysis.

Among these 33 patients, the median age at inclusion was 2 years (IQR 2.0–5.0). The median glomerular filtration rates were 115 (104–128) mL/min/1.73 $m^2$ (**Table 1**). The median time between NS diagnosis and renal biopsy was 2.1 years (IQR 0.9–3.9). Ten (30.3%) patients were corticosteroid resistant. The most frequent pathology diagnosed by biopsy was

**Table 1. Demographic and clinical characteristics of patients with NS at diagnosis.**

| Characteristics | N = 33 | |
|---|---|---|
| | **n** | **%** |
| **Age at diagnosis [years]**[*] | 2.0 (2.0–5.0) | |
| **Sex** | | |
| Female | 11 | 33,3 |
| Male | 22 | 66,7 |
| **Ethnicity** | | |
| Mestizo | 24 | 72,7 |
| Indigenous | 1 | 3,0 |
| Black | 8 | 24,2 |
| **Siblings with NS** | 5 | 15,2 |
| **Variables at diagnosis** | | 0,0 |
| Macroscopic hematuria | 2 | 6,1 |
| Microscopic hematuria | 8 | 24,2 |
| Arterial hypertension | 11 | 33,3 |
| GFR [ml/min/1.73 $m^2$][*] | 115 (104–128) | |
| **KDIGO classification at diagnosis** | | |
| 1 | 29 | 87,9 |
| 2 | 3 | 9,1 |
| 3 | 1 | 3,0 |
| **Initial treatment** | | |
| Corticosteroids | 30 | 90,9 |
| Mycophenolate mofetil | 1 | 3,0 |
| Tacrolimus | 1 | 3,0 |
| Rituximab | 2 | 6,1 |
| ACE inhibitor | 16 | 48,5 |
| Cyclophosphamide | 4 | 12,1 |
| Cyclosporine | 3 | 9,1 |

[*]Median (IQR). GFR, glomerular filtration rate. ACE, angiotensin converting enzyme.

**Table 2. Clinical characteristics of patients with NS at the time of renal biopsy.**

| Characteristics | N = 33 | |
|---|---|---|
| | **n** | **%** |
| **Age at biopsy [years]**[*] | 6.5 (4.3–9.3) | |
| **Time between diagnosis and biopsy [years]**[*] | 2.1 (0.9–3.9) | |
| **GFR [ml/min/1.73 m²]**[*] | 108.4 (73.5–134.1) | |
| **KDIGO classification at biopsy** | | |
| 1 | 22 | 66,7 |
| 2 | 7 | 21,2 |
| 3 | 2 | 6,1 |
| 4 | 1 | 3,0 |
| 5 | 1 | 3,0 |
| **Variables at biopsy** | | |
| Macroscopic hematuria | 3 | 9,1 |
| Microscopic hematuria | 8 | 24,2 |
| Arterial hypertension | 6 | 18,2 |
| **Response to corticosteroids** | | |
| Corticosteroid-responsive | 8 | 24,2 |
| Corticosteroid-dependent | 12 | 36,4 |
| Corticosteroid-resistant | 10 | 30,3 |
| No data | 3 | 9,1 |
| **Treatment at biopsy** | | |
| Cyclophosphamide | 9 | 27,3 |
| Cyclosporine | 9 | 27,3 |
| ACE inhibitor | 28 | 84,8 |
| **Nephrotic proteinuria** | 21 | 63,6 |
| No data | 2 | 6,1 |
| **History of renal failure** | 10 | 30,3 |
| No data | 1 | 3,0 |
| **Drug-induced nephrotoxicity** | 4 | 12,1 |
| No data | 2 | 6,1 |
| **Biopsy findings** | | |
| Minimal change nephropathy | 21 | 63.6 |
| Focal and segmental glomerulosclerosis | 12 | 36.4 |

[*]Median (IQR). GFR, glomerular filtration rate. ACE, angiotensin converting enzyme.

MCD, which was found in 21 (63.6%) patients, followed by focal segmental glomerulosclerosis, which was found in 12 (36.4%) patients. **Table 2** describes the clinical characteristics at the time of renal biopsy, and **Table 3** describes the same characteristics at the last follow-up.

## BAFF and receptor expression in renal biopsies

To evaluate the renal distribution of BAFF and its three receptors, we performed immunohistochemistry on formalin-fixed, paraffin-embedded renal biopsies of the 33 included patients. The expression of the different markers was evaluated on podocytes, III, and parietal cells. **Table 4** shows the different expression patterns.

Podocytes presented expression of the BAFF, TACI, and BCMA in several patients, while III presented expression of the 4 markers in at least one patient. On the other hand, parietal cells did not show any expression of the evaluated markers.

**Table 3. Clinical characteristics of patients with NS at the last follow-up.**

| Characteristics | | n = 33 | |
|---|---|---|---|
| | | **n** | **%** |
| **Age at last follow-up [years]***  | | 11.3 (7.5–14.4) | |
| **Time between biopsy and last follow-up [years]***  | | 4.2 (1.6–5.6) | |
| **GFR [ml/min/1.73 m²]***  | | 110 (83–110) | |
| **KDIGO classification at last follow-up** | | | |
| | 1 | 21 | 63,6 |
| | 2 | 10 | 30,4 |
| | 4 | 1 | 3,0 |
| | 5 | 1 | 3,0 |
| **Variables at last follow-up** | | | |
| Macroscopic hematuria | | 0 | 0,0 |
| Microscopic hematuria | | 6 | 18,2 |
| Arterial hypertension | | 2 | 6,1 |
| Edema | | 1 | 3,0 |
| **Treatment at last follow-up** | | | |
| Corticosteroids | | 22 | 66,7 |
| Rituximab | | 0 | 0,0 |
| ACE inhibitor | | 22 | 66,7 |
| Cyclophosphamide | | 9 | 27,3 |
| Cyclosporine | | 12 | 36,4 |
| **Drug-induced nephrotoxicity** | | 1 | 3,0 |
| **History of renal failure** | | 6 | 18,2 |
| **Renal replacement therapy** | | 6 | 18,2 |
| Type of renal replacement therapy | | | |
| Hemodialysis | | 3 | 9,1 |
| Peritoneal dialysis | | 2 | 6,1 |
| **Kidney transplant** | | 1 | 3,0 |
| **Death** | | 1 | 3,0 |

*Median (IQR). GFR, glomerular filtration rate. ACE, angiotensin converting enzyme.

**BAFF** was the most common marker seen in renal samples. It was found in both podocytes (18.2% of samples) and III (36.4% of samples). BAFF expression was predominantly slight in most cases.

On the other hand, the receptor expression was variable, with BAFF-R found in only one sample (in the III), TACI in 4 samples (in both podocytes and III) and BCMA in 5 samples in the podocytes and 7 cases in the IIIs. **Fig 1** illustrates BAFF and TACI expression in renal podocytes and III.

## BAFF expression is associated with the worst renal outcomes

Kaplan-Meier curves were generated to analyze renal function defined as renal failure when GFR was $\leq$ 90 mL/min/1.73 m² at 60 months of follow-up. At this time, 69.6% of the patients with MCD and 80.1% of the patients with FSGS maintained a GFR $\geq$ 90 mL/min/1.73 m² (p = 0.4513).

After a median follow-up of 60 months, the probability of retaining a normal eGFR is or is not significantly different between groups expressing BAFF in podocytes and III. Patients with BAFF expression in renal podocytes had a 25% probability of having a GFR > 90 mL/min/1.73 m²; instead, patients with BAFF absence had a 84.9% probability of having a GFR>90 mL/

**Table 4. Biopsy biomarkers in patients with NS, N = 33.**

| Biomarker | Podocytes | | Interstitial inflammatory infiltrate | |
|---|---|---|---|---|
| | n | % | n | % |
| **BAFF** | | | | |
| Negative | 27 | 81,8 | 21 | 63,6 |
| Positive | 6 | 18,2 | 12 | 36,4 |
| **Extent\*** | | | | |
| 0–25% | 6 | 100 | 8 | 66,6 |
| 26–50% | 0 | 0 | 2 | 16,7 |
| ≥51% | 0 | 0 | 2 | 16,7 |
| **BAFF-R** | | | | |
| Negative | 22 | 0 | 32 | 97 |
| Positive | 0 | 0 | 1 | 3 |
| **Extent\*** | | | | |
| 0–25% | – | – | 0 | – |
| 26–50% | – | – | 1 | 100 |
| ≥51% | – | – | 0 | – |
| **TACI** | | | | |
| Negative | 29 | 87,9 | 29 | 87,9 |
| Positive | 4 | 12,1 | 4 | 12,1 |
| **Extent\*** | | | | |
| 0–25% | 4 | 100 | 3 | 75 |
| 26–50% | 0 | 0 | 0 | 0 |
| ≥51% | 0 | 0 | 1 | 25 |
| **BCMA** | | | | |
| Negative | 28 | 84,8 | 26 | 78,8 |
| Positive | 5 | 15,2 | 7 | 21,2 |
| **Extent\*** | | | | |
| 0–25% | 4 | 80 | 6 | 85,7 |
| 26–50% | 1 | 20 | 1 | 14,3 |
| ≥51% | 0 | 0 | 0 | 0 |

\*Percentages based on positive values. None of the biomarkers were expressed in parietal cells.

min/1.73 m$^2$ (*p* = 0.0067) (**Fig 2**). Patients with BAFF present in renal III had a 42.9% probability of having GFR>90 mL/min/1.73 m$^2$, while in patients with the absence of BAFF in this tissue, the probability was 94.1% (*p* = 0.0063) (**Fig 3**). These results show that if a pediatric patient with NS expresses BAFF either on the podocytes or the III, there is an increased possibility of worse renal function at 60 months following the renal sample evaluation.

On the other hand, patients with BCMA (one of the three receptors for BAFF that was evaluated) expression in renal III had a 53.3% probability of having GFR>90 mL/min/1.73 m$^2$, while in patients with the absence of BCMA in this tissue, the probability was 79.3% (*p* = 0.3814) (**Fig 4**).

Although TACI and BAFF-R were found in some patients, the low number of events prevented survival analysis of renal function (**Table 4**).

## Renal outcomes according to response to corticosteroids

A Kaplan-Meier curve was generated to analyze renal function defined as renal failure when GFR was ≤ 90 mL/min/1.73 m$^2$ at 60 months of follow-up according to steroid response. No

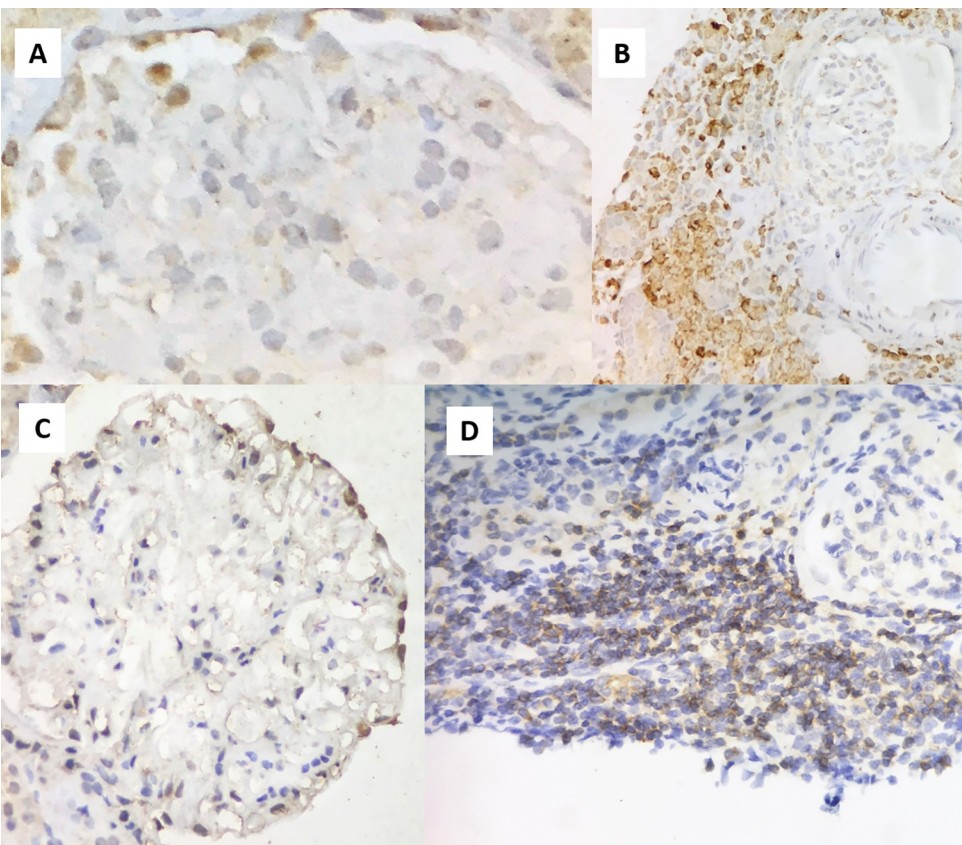

**Fig 1. BAFF and TACI expression in renal tissue: 1A) BAFF is expressed in the glomerular podocytes. 1B)** BAFF expression in the interstitial inflammatory infiltrate. **1C)** TACI expression in the glomerular podocytes. **1D)** TACI expression in the interstitial inflammatory infiltrate.

differences were found between steroid responders (78.1% of probability) and steroid resistants (72.9%) (***p* = 0.9718**) (**Fig 5**).

## Discussion

The cause of idiopathic NS remains unknown, but the immune role of B and T cells in the pathogenesis of this condition has been suggested. Although depleting B cells with rituximab (anti-CD-20 therapy) is an effective treatment in patients with a complicated disease [10], the mechanisms involving B cells in patients with NS are not well described. In addition, the role of BAFF (an important cytokine associated with B-cell survival and autoimmunity) has not been studied in pediatric patients with NS.

Our study, including 33 pediatric patients with NS in whom a renal biopsy was performed, showed that the expression of BAFF on podocytes and III was associated with the worst renal outcome at 60 months. These results highlight the importance of local B-cell survival factors such as BAFF in the pathogenesis and prognosis of pediatric patients with NS.

The value of this study lies in it being the first description of this marker in kidney samples of patients with NS and of the long-term reduction in kidney function in patients expressing the marker. Even more interestingly, targeting BAFF therapies (i.e., belimumab) are available and approved for autoimmune disorders characterized by BAFF overproduction, mainly SLE [15] and recently in patients with lupus nephritis [16] as has been demonstrated by methods of

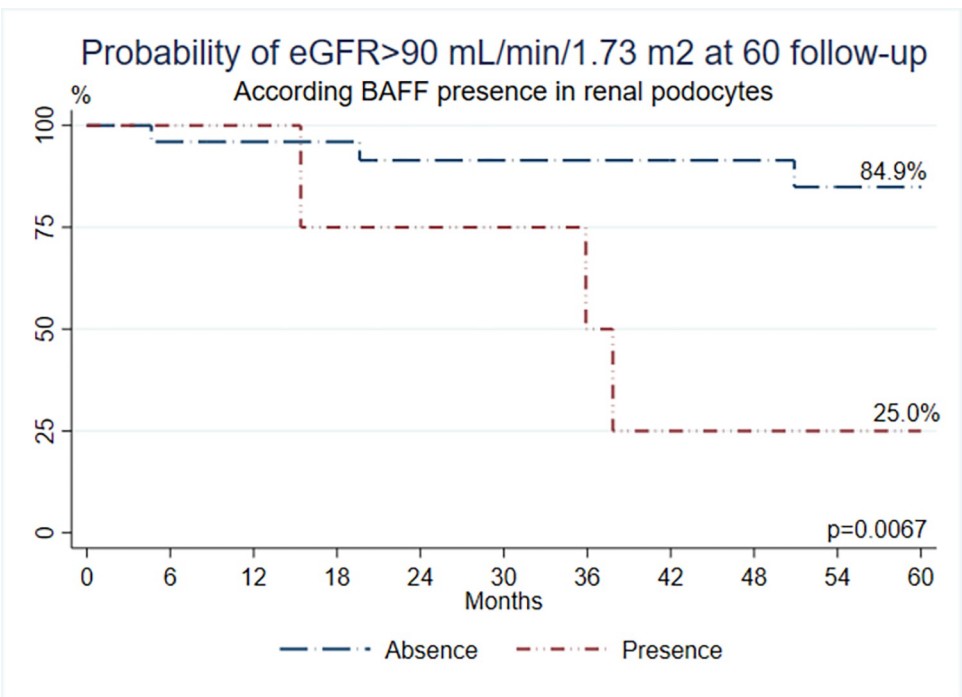

**Fig 2. Probability of eGFR >90mL/min/1.73m2 at 60 months follow-up according to BAFF expression in renal podocytes.**

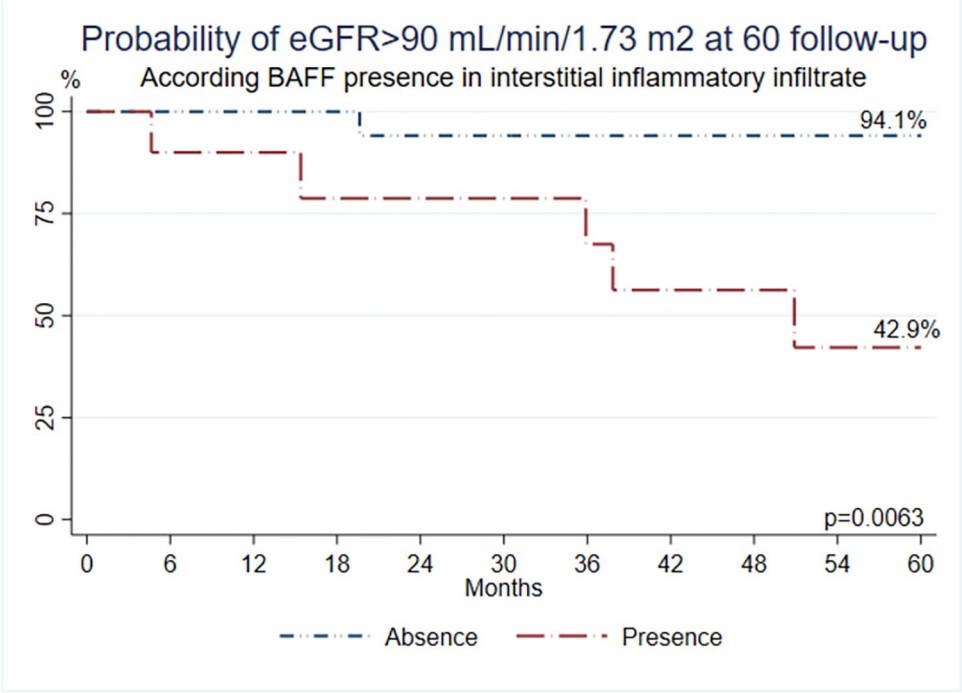

**Fig 3. Probability of eGFR >90mL/min/1.73m2 at 60 months follow-up according to BAFF expression in renal interstitial inflammatory infiltrate.**

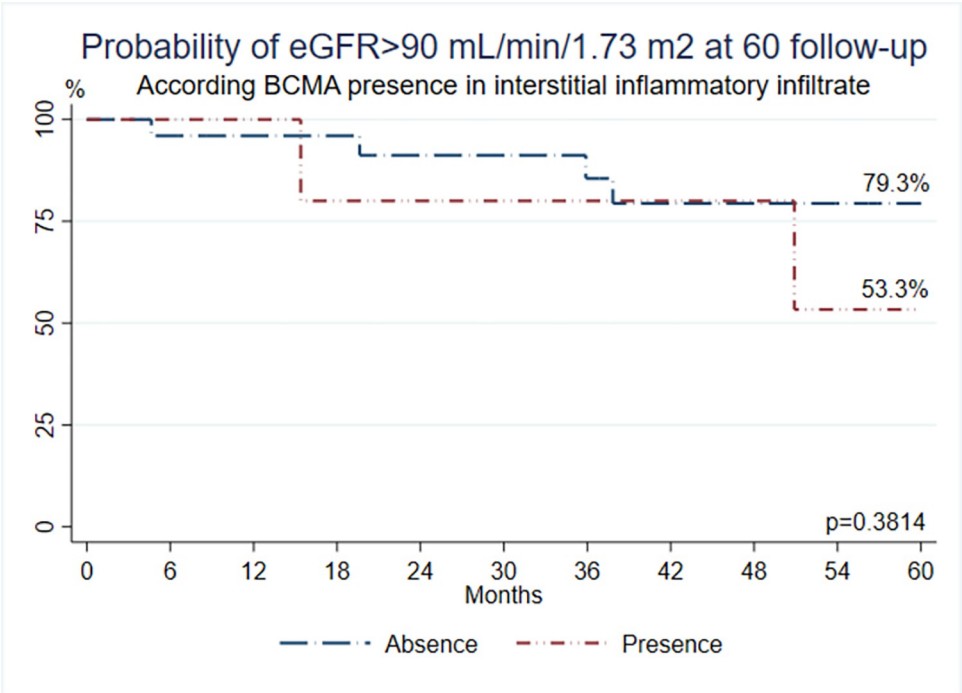

**Fig 4. Probability of eGFR >90mL/min/1.73m2 at 60 months follow-up according to BCMA expression in renal interstitial inflammatory infiltrate.**

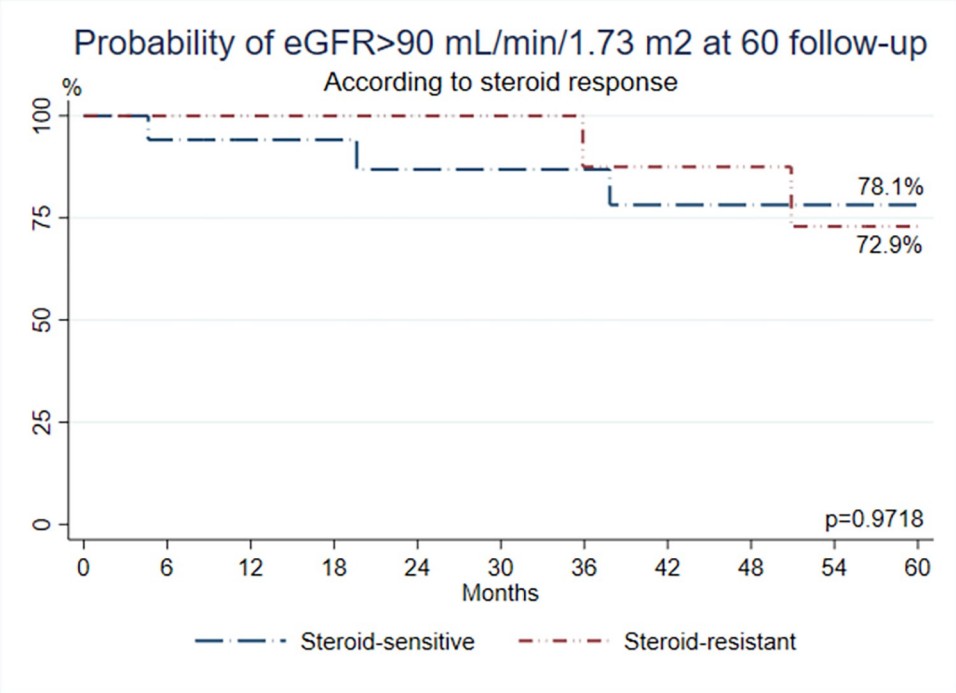

**Fig 5. Probability of eGFR >90mL/min/1.73m2 at 60 months follow-up according to steroid response.**

immunohistochemistry [14] and RNAseq [17] of renal biopsies, suggesting that giving anti-BAFF drugs may be a future strategy to treat patients with NS. Furthermore, belimumab has shown effectiveness on reducing anti-phospholipase A2 receptor autoantibodies and protein-uria in in primary membranous nephropathy [18].

In our study, all 3 BAFF receptors (BCMA, TACI and BAFF-R) were found in some samples, with BCMA being the most common receptor found in 36.4% of cases. Although the extent of the expressed biomarkers was generally low, we observed that they were positive in different locations (including podocytes and III) but not in parietal cells.

Concerning the involvement of BAFF and its receptors in kidney-mediated diseases, we have previously demonstrated that the expression patterns of BAFF and its receptors differ according to **lupus nephritis** class in adult patients, where it is more important in Class IV with high activity indexes [14]. These results, as well as the results obtained from other groups, were the basis of evaluating the efficacy of BAFF-targeting therapies in lupus nephritis [16]. In pediatric lupus patients, Edelbauer *et al.* [13] showed that elevated plasmatic levels of BAFF were predictors of relapsing disease. However, our present study is the first to show the expression of BAFF as a histopathological marker of an autoimmune disease in children other than SLE.

The clinical use of BAFF as a biomarker in nephrology has gained interest in other conditions, such as renal transplantation and rejection.

For instance, Thibault-Espitia *et al.* [19] analyzed the levels of BAFF mRNA and its soluble protein, as well as transcripts coding for its receptors, namely, BAFF-R, TACI and BCMA, in the blood of 143 patients with stable kidney transplant function 5 years or more posttransplantation. They showed that stable patients with high BAFF-R levels had a higher risk of developing graft dysfunction, and patients with lower levels of BAFF transcripts or a higher level of soluble BAFF had a significantly higher risk of developing donor-specific antibodies, suggesting that BAFF may be used as a risk factor for renal graft dysfunction and the development of donor-specific antibodies.

More recently, Wang *et al.* in China [20] confirmed the hypothesis that BAFF may be a marker to predict acute rejection in kidney transplant patients. Serum BAFF levels when acute rejection occurred were significantly higher than those in the stable renal function group ($P < 0.05$). BAFF expression was significantly enhanced in the membrane and cytoplasm of renal tubule epithelial cells in the transplant kidney tissue with acute rejection, showing a positive correlation between BAFF and C4d expression ($r = 0.880$, $P = 0.001$). In this study, Kaplan-Meier survival analysis showed that recipients with higher pretransplant BAFF levels had higher acute rejection incidence ($P = 0.003$). All these data suggest that BAFF levels are associated with acute rejection and could be a promising biomarker to predict kidney transplant rejection risks.

The strength of our study lies in the description of an unknown immunological component in NS pathogenesis, giving an important role to B lymphocytes in the pathogenesis. The source and regulation of BAFF in the kidney tissue of patients with NS need to be defined to better understand the immune dysregulation seen in this clinical condition.

The finding of these markers in the renal tissue of patients with NS and the association of BAFF expression with the worst renal prognosis also opens the door for the possible use of therapeutic strategies against BAFF (i.e., belimumab), which has demonstrated benefits in patients with SLE and lupus nephritis [16]. For instance, the use of belimumab showed positive results in kidney function, proteinuria and GFR, with a similar safety profile to conventional therapies.

In conclusion, our findings suggest the involvement of BAFF and its receptors in the immune response in pediatric patients with NS. In addition, this is the first study to find a

relationship between the presence of BAFF and long-term reduction of renal function in pediatric patients with NS. According to our results, the presence of BAFF in renal podocytes and renal III could act as a prognostic factor for renal function. Prospective studies are required to confirm this hypothesis.

## Study limitations

Given the nature of our study, access to medical information was restricted to the available medical charts in our center. Although there are others (e.g. RNAseq), a single research methodology (immunohistochemistry) was performed; it could be useful to combine methods in order to have a wider perspective of the biomarkers presence in tissue.

## Acknowledgments

Thank you to Centro de Investigaciones Clínicas of Fundación Valle del Lili for the support.

## Author Contributions

**Conceptualization:** Jaime M. Restrepo, María Claudia Barrera, Gabriel J. Tobón.

**Data curation:** Jessica Forero-Delgadillo, Vanessa Ochoa, Jaime M. Restrepo, Laura Torres-Canchala, Ivana Nieto-Aristizábal, Ingrid Ruiz-Ordoñez, Carlos Andrés Jimenez.

**Formal analysis:** Jessica Forero-Delgadillo, Vanessa Ochoa, Jaime M. Restrepo, Laura Torres-Canchala, Carlos Andrés Jimenez, Gabriel J. Tobón.

**Funding acquisition:** Jaime M. Restrepo.

**Investigation:** Jaime M. Restrepo, Laura Torres-Canchala, Ivana Nieto-Aristizábal, Ingrid Ruiz-Ordoñez, Carlos Andrés Jimenez, Gabriel J. Tobón.

**Methodology:** Laura Torres-Canchala, Aura Sánchez, María Claudia Barrera, Carlos Andrés Jimenez, Gabriel J. Tobón.

**Project administration:** Jaime M. Restrepo.

**Resources:** Jaime M. Restrepo, Aura Sánchez, Carlos Andrés Jimenez.

**Supervision:** Jaime M. Restrepo, Gabriel J. Tobón.

**Validation:** Jaime M. Restrepo.

**Writing – original draft:** Jessica Forero-Delgadillo, Vanessa Ochoa, Jaime M. Restrepo, Laura Torres-Canchala, Ivana Nieto-Aristizábal, Ingrid Ruiz-Ordoñez, Carlos Andrés Jimenez, Gabriel J. Tobón.

**Writing – review & editing:** Jessica Forero-Delgadillo, Vanessa Ochoa, Jaime M. Restrepo, Laura Torres-Canchala, Ivana Nieto-Aristizábal, Ingrid Ruiz-Ordoñez, Aura Sánchez, María Claudia Barrera, Carlos Andrés Jimenez, Gabriel J. Tobón.

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
