## [Decision Letter · Decision Letter 0]

24 Jun 2022

PONE-D-22-07630B-cell activating factor (BAFF) and its receptors’ expression in pediatric nephrotic syndrome is associated with worse prognosisPLOS ONE

Dear Dr. Tobón,

Thank you for submitting your manuscript to PLOS ONE. After careful consideration, we feel that it has merit but does not fully meet PLOS ONE’s publication criteria as it currently stands. Therefore, we invite you to submit a revised version of the manuscript that addresses the points raised during the review process.

ACADEMIC EDITOR: This study seems to be of interest to the paediatric nephrology community, but would require major revisions that are clearly outlined by two expert reviewers, that in my view should be surmountable. Pls do take these recommendations at heart, especially on the scientific rationale of the study and hypothesis. Given the small sample size (which is understandable in paediatric nephrology) always warrants caution especially given heterogeneity within the sample size, too. Specific questions from reviewers need to be addressed in a point-by-point fashion, and their recommendations followed in great detail. Please be cautious using strong statements as conclusions given the point made re: methodology and sample size/heterogeneity.

 Please submit your revised manuscript by Aug 08 2022 11:59PM. If you will need more time than this to complete your revisions, please reply to this message or contact the journal office at plosone@plos.org. Please include the following items when submitting your revised manuscript:A rebuttal letter that responds to each point raised by the academic editor and reviewer(s). You should upload this letter as a separate file labeled 'Response to Reviewers'.A marked-up copy of your manuscript that highlights changes made to the original version. You should upload this as a separate file labeled 'Revised Manuscript with Track Changes'.An unmarked version of your revised paper without tracked changes. You should upload this as a separate file labeled 'Manuscript'.

We look forward to receiving your revised manuscript.

Kind regards,

Frank JMF Dor, M.D., Ph.D., FEBS, FRCS

Academic Editor

PLOS ONE

Journal Requirements:

Reviewers' comments:

Reviewer's Responses to Questions

**Comments to the Author**

1. Is the manuscript technically sound, and do the data support the conclusions?

Reviewer #1: Yes

Reviewer #2: Partly

2. Has the statistical analysis been performed appropriately and rigorously? 

Reviewer #1: Yes

Reviewer #2: Yes

3. Have the authors made all data underlying the findings in their manuscript fully available?

Reviewer #1: Yes

Reviewer #2: Yes

4. Is the manuscript presented in an intelligible fashion and written in standard English?

Reviewer #1: Yes

Reviewer #2: No

5. Review Comments to the Author

Reviewer #1: Congratulations to authors on a well written manuscript.It addresses very important topic of an important childhood disease and provides decent literature overview.

My comments and recommendations are:

1.Please reference paragraph 2 in Introduction section (lines 93-96).

2. Was there a reason not to comment on cyclophosphamide as treatment for NS? (lines 106-108). I recommend that all immunosuppressive agents are addressed adequately in order to avoid bias.

3. Expand on definition of frequent relapse (lines 150-151). You have used only one reference to make a strong statement. I suggest you add more data to support this statement.

4. Further clarification and referencing is needed in Statistical analysis section: how did you decide to to such clarification such as 0-25% as one group, etc. This also needs a reference.

5. Can you further define group in which the BAFF expression was positive? What treatment were those patients initially on? Were they MCD or FSGS?

6. Based on your findings, can you provide recommendations (ie should all kidney biopsies be stained for BAFF expression? What, if any, approved for use in paediatrics drugs would you advise in those patients?)

Reviewer #2: The authors investigated the expression of BAFF and related receptors (BAFF-R, BCMA and TACI) in paediatric nephrotic syndrome, and concluded that BAFF expression in kidney was associated with less favourable outcomes. My comments and concerns are as follows:

1. The scientific rationale and hypothesis of this study is not strong; and is not clearly explained in the Introduction.

2. The sample size is relatively small, and the patient characteristics are heterogenous.

3. The overall results are mixed and hence no definition conclusion can be drawn.

4. The choice of outcome (eGFR >90) is rather unorthodox; more conventional outcomes should be remission status (e.g. CR or PR or NR), development of CKD, doubling of Scr or ESKD.

5. Table 4 should be tabulated according to the underlying renal pathology (MCD vs. FSGS).

6. There are many typos in Figure 2&3 (e.g. absence, at 60 months; according to, etc).

6. PLOS authors have the option to publish the peer review history of their article (what does this mean?). If published, this will include your full peer review and any attached files.

Reviewer #1: **Yes: **Jelena Stojanovic

Reviewer #2: No

---

## [Author Response · Author response to Decision Letter 0]

9 Aug 2022

Aug 04, 2022

PLOS ONE

Re: BAFF and its receptors’ expression in pediatric nephrotic syndrome is associated with a worse prognosis

Dear Editor

Thank you for reviewing our manuscript and for the reviewers’ comments. We have addressed the concerns. Please find our comments and responses to the concerns below. For added clarity about the point-to-point reply, we have first repeated the requests in normal font, added our comments in italics, and changes to the manuscript were indicated in highlighted text. We hope that the edited manuscript is now acceptable for publication and wish to resubmit it.

Reviewer 1

1 .Please reference paragraph 2 in Introduction section (lines 93-96).

Answer: We added some references to the paragraph. 

“The most common causes of NS in pediatric patients are minimal change disease (MCD) in up to 90% of cases (associated with good prognosis) [1] and focal segmental glomerulosclerosis (FSGS) in 10-15% of cases [2]. The latter is associated with a worse prognosis and risk of progression to chronic kidney disease (CKD) [3].

2. Was there a reason not to comment on cyclophosphamide as a treatment for NS? (lines 106-108). I recommend that all immunosuppressive agents are addressed adequately in order to avoid bias.

Answer: We thank the reviewer's comment. Most NS patients, around 80%, can be treated only with steroids. In Steroid-Resistant Nephrotic Syndrome (SRNS) patients, we add immunosuppressor agents. Of our patients, 12 were SRNS; of them, five patients received cyclophosphamide; 2 patients, cyclosporine; 1 patient, mycophenolate mofetil; 2 patients, rituximab; 1 patient, tacrolimus and 1 patient cyclophosphamide six-months regimen and then was changed to cyclosporine due to therapeutic failure. We decided to add this last patient to the cyclosporine group. According to the reviewer's comment, we divided SRSN patients into two groups: cyclophosphamide treated patients (n=5) and other immunosuppressors treated patients (n=7). The median eGFR in cyclophosphamide group was 83 mL/min/1.73m2 (RIC 67-120) and other immunosuppressors treated patients eGFR was 110 mL/min/1.73m2 (RIC 79-115) with no significant differences (p=0.9353). We modified table 1 and added the next paragraph below.

Methods section

Among these 33 patients, the median age at inclusion was 2 years (IQR 2.0 – 5.0). The median glomerular filtration rates were 115 (104-128) mL/min/1.73 m2 (Table 1). The median time between NS diagnosis and renal biopsy was 2.1 years (IQR 0.9-3.9). Twelve (36.6%) patients were Steroid-Resistant Nephrotic Syndrome (SRNS) patients. Of them 5/12 patients received cyclophosphamide and 7/12, other immunosuppressor treatment. The median eGFR in cyclophosphamide group was 83 mL/min/1.73m2 (RIC 67-120) and other immunosuppressors group, eGFR was 110 mL/min/1.73m2 (RIC 79-115) with no significant differences (p=0.9353). The most frequent pathology diagnosed by biopsy was MCD, which was found in 21 (63.6%) patients, followed by focal segmental glomerulosclerosis, which was found in 12 (36.4%) patients. Table 2 describes the clinical characteristics at the time of renal biopsy, and Table 3 describes the same characteristics at the last follow-up.

Characteristics N=33

 n %

Age at diagnosis [years]* 2.0 (2.0-5.0)

Sex 

Female 11 33,3

Male 22 66,7

Ethnicity 

Mestizo 24 72,7

Indigenous 1 3,0

Black 8 24,2

Siblings with NS 5 15,2

Variables at diagnosis 0,0

Macroscopic hematuria 2 6,1

Microscopic hematuria 8 24,2

Arterial hypertension 11 33,3

GFR [ml/min/1.73 m2]* 115 (104-128)

KDIGO classification at diagnosis 

1 29 87,9

2 3 9,1

3 1 3,0

Initial treatment 

Corticosteroids 30 90,9

Mycophenolate mofetil 1 3,0

Tacrolimus 1 3,0

Rituximab 2 6,1

ACE inhibitor 16 48,5

Cyclophosphamide 5 15,5

Cyclosporine 3 9,1

3. Expand on the definition of frequent relapse (lines 150-151). You have used only one reference to make a strong statement. I suggest you add more data to support this statement.

Answer: We included the following reference “Lancet. 2018 Jul 7;392(10141):61-74. doi: 10.1016/S0140-6736(18)30536-1" which defines frequent relapse as ≥2 relapses within 6 months of initial response or ≥4 in any 12-month period, similar to the definition we provided in the Methods´ section (≥ 4 relapses in any 12-month period). 

The reference was included in the reference list. 

“Methods 

…frequent relapse (≥ 4 relapses in any 12-month period) [4].

4. Further clarification and referencing are needed in the Statistical analysis section: how did you decide to do such clarification such as 0-25% as one group, etc. 

a reference.

Answer: We considered that the most objective way to measure our findings regarding the glomerular and interstitial biomarkers’ expression was to extrapolate the method from an already stablished one as the one from the International Society of Nephrology/Renal Pathology Society for the lupus nephritis classification (Kidney International (2018) 93, 789–796; https://doi.org/10.1016/ j.kint.2017.11.023). This was clarified on methods with the supporting reference. 

“Statistical analysis 

Slides were read by one expert nephropathologist blinded to clinical outcome, and samples were scored according to the percentage of cells with positive expression on podocytes and interstitial inflammatory infiltrates (III) (mononuclear cells, B or T-cells), as follows: 1) 1-25% of cells stained; 2) 26-50% of cells stained; and 3) more than 50% of cells stained (extrapolated from the International Society of Nephrology/Renal Pathology Society classification for lupus nephritis [5] .

 5. Can you further define the group in which the BAFF expression was positive? What treatment were those patients initially on? Were they MCD or FSGS?

Answer. We thank the reviewer comment. Unfortunately, due to the low needing of biopsy in nephrotic syndrome, our sample size is low. It does not allow us to draw precise conclusions. That is why we have decided not to add the tables in the article. However, we present the tables disaggregated by type of disease below.

Minimal Changes FSGS

Biomarker Podocytes Interstitial inflammatory infiltrate Biomarker Podocytes Interstitial inflammatory infiltrate

 n % n % n % n %

BAFF BAFF 

Negative 17 80 14 67 Negative 10 83.3 7 58.3

Positive 4 19 7 33 Positive 2 16.7 5 41.7

Extent* Extent* 

0-25% 4 100 5 71.4 0-25% 2 100 3 60

26-50% 0 0 1 14.2 26-50% 0 0 1 10

≥51% 0 0 1 14.2 ≥51% 0 0 1 10

BAFF-R BAFF-R 

Negative 21 100 21 100 Negative 0 0 11 91.7

Positive 0 0 0 0 Positive 0 0 1 8.3

Extent* Extent* 

0-25% 0 0 0 0 0-25% 0 0 0 0

26-50% 0 0 0 0 26-50% 0 0 1 100

≥51% 0 0 0 0 ≥51% 0 0 0 0

TACI 0 0 0 0 TACI 

Negative 19 91.5 19 91.5 Negative 10 83.3 10 83.3

Positive 2 9.5 2 9.5 Positive 2 16.7 2 16.7

Extent* Extent* 

0-25% 2 100 2 100 0-25% 2 100 1 50

26-50% 0 0 0 0 26-50% 0 0 0 0

≥51% 0 0 0 0 ≥51% 0 0 1 50

BCMA BCMA 

Negative 18 85.7 17 81 Negative 10 83.3 9 75

Positive 3 14.3 4 19 Positive 2 16.7 3 25

Extent* Extent* 

0-25% 3 100 4 100 0-25% 10 83.4 2 66.7

26-50% 0 0 0 0 26-50% 1 8.3 1 33.3

≥51% 0 0 0 0 ≥51% 1 8.3 0 0

6. Based on your findings, can you provide recommendations (ie should all kidney biopsies be stained for BAFF expression? What, if any, approved for use in pediatrics drugs would you advise in those patients?)

Answer: Nephrotic syndrome is a disease in which immunological mechanisms are being studied. In case that BAFF and its receptors become stablished as routine biomarkers, they would be helpful in the follow-up of patients with perhaps, worst prognosis. Additionally, with the approval of anti-BAFF drugs for children, a possibility of treatment can be purposed. To make this clear in the manuscript, a reference with the FDA approval for Belimumab in children was added in the third paragraph of the discussion and a final paragraph was included to state possible recommendations from our point of view based on our findings. 

“Discussion (third paragraph)

The value of this study lies in it being the first description of this marker in kidney samples of patients with NS and of the long-term reduction in kidney function in patients expressing the marker. Even more interestingly, targeting BAFF therapies (i.e., belimumab) are available and approved by the Food and Drug Administration for autoimmune disorders characterized by BAFF overproduction, mainly SLE [6] and recently in patients with lupus nephritis [16] as has been demonstrated by methods of immunohistochemistry[14] and RNAseq [17] of renal biopsies, suggesting that giving anti-BAFF drugs may be a future strategy to treat patients with NS. Furthermore, belimumab has shown effectiveness on reducing anti-phospholipase A2 receptor autoantibodies and proteinuria in in primary membranous nephropathy [18].”

“Discussion (last paragraph) 

In this way, NS is a disease in which immunological factors as BAFF and it receptors are being described, so an interest to evaluate them in the clinical practice should arise as they could imply a worst prognosis or the need for a closer follow-up in these patients. Furthermore, the availability of anti-BAFF therapies approved for pediatric patients state a possibility of treatment for patients who present those biomarkers. Prospective studies are required to confirm this hypothesis.”

Reviewer 2

The authors investigated the expression of BAFF and related receptors (BAFF-R, BCMA, and TACI) in pediatric nephrotic syndrome, and concluded that BAFF expression in the kidney was associated with less favorable outcomes. My comments and concerns are as follows:

The scientific rationale and hypothesis of this study are not strong and are not clearly explained in the Introduction.

Answer: To state this clearer, we modified the last paragraph of the Introduction as follows:

 “Introduction (last paragraph)

With the aim to provide further knowledge on any possible immunopathogenic mechanism involved in NS and as the role of BAFF and its receptors in pediatric patients with NS has not been evaluated, we investigated the expression of BAFF and its three receptors in kidney biopsies from patients with NS and to correlate their expression with clinical outcomes. The identification of BAFF and its receptors in renal biopsies from NS could lead to their application in the clinical practice as potential biomarkers and as a possible target of immunomodulatory therapies.”

2. The sample size is relatively small, and the patient characteristics are heterogeneous.

Answer: We thank the reviewer's comments. Unfortunately, given the low frequency of the need for biopsy in patients with nephrotic syndrome, our sample is limited. This means that the sample size is not probabilistic. We add this observation in the study limitation section.

STUDY LIMITATIONS:

Given the nature of our study, access to medical information was restricted to the available medical charts in our center. Because the renal biopsy needing in patients with nephrotic syndrome is low (the management can be done without the biopsy in most cases), our sample size is small.

3. The overall results are mixed and hence no definition conclusion can be drawn.

Answer: To state our conclusions clearer, we complemented the discussion as follows:

“Discussion (third paragraph)

The value of this study lies in it being the first description of this marker in kidney samples of patients with NS and of the long-term reduction in kidney function in patients expressing the marker. Even more interestingly, targeting BAFF therapies (i.e., belimumab) are available and approved by the Food and Drug Administration for autoimmune disorders characterized by BAFF overproduction, mainly SLE [15] (for adults since 2011, and children since 2019 FDA approves first treatment for pediatric patients with lupus | FDA )and recently in patients with lupus nephritis [16] as has been demonstrated by methods of immunohistochemistry[14] and RNAseq [17] of renal biopsies, suggesting that giving anti-BAFF drugs may be a future strategy to treat patients with NS. Furthermore, belimumab has shown effectiveness on reducing anti-phospholipase A2 receptor autoantibodies and proteinuria in in primary membranous nephropathy [18].”

“Discussion (last paragraph) 

In this way, NS is a disease in which immunological factors as BAFF and it receptors are being described, so an interest to evaluate them in the clinical practice should arise as they could imply a worst prognosis or the need for a closer follow-up in these patients. Furthermore, the availability of anti-BAFF therapies approved for pediatric patients state a possibility of treatment for patients who present those biomarkers. Prospective studies are required to confirm this hypothesis.”

4. The choice of outcome (eGFR >90) is rather unorthodox; more conventional outcomes should be remission status (e.g. CR or PR or NR), development of CKD, doubling of Scr or ESKD.

Answer: We appreciate the reviewer's comment. Although chronic kidney failure (eGFR <60mL/min/1.73m2 [KDIGO stage III or more]) is a relevant outcome to study, our objective in this study was to detect reduced kidney function, not chronic kidney failure. Also, fortunately only one of our patients had eGFR stage IV according to KDIGO, so it was impossible to perform an analysis using eGFR < 60mL/min/1.73m2 cut-off point.

The renal function according to steroid response was already compared in the manuscript. We present the paragraph below

Results

Renal outcomes according to response to corticosteroids 

A Kaplan-Meier curve was generated to analyze renal function defined as renal failure when GFR was ≤ 90 mL/min/1.73 m2 at 60 months of follow-up according to steroid response. No differences were found between steroid responders (78.1% of probability) and steroid resistants (72.9%) (p=0.9718) (Figure 5).

Figure 5. Probability of eGFR >90mL/min/1.73m2 at 60 months follow-up according to steroid response. 

5.Table 4 should be tabulated according to the underlying renal pathology (MCD vs. FSGS).

Answer. We thank the reviewer comment. The requested changes have already been made according to the reviewer 1 observation.

6. There are many typos in Figures 2&3 (e.g. absence, at 60 months; according to, etc).

Answer. We thank the reviewer comment. The figures were modified as follow

Figure 2. Probability of eGFR >90mL/min/1.73m2 at 60 months follow-up according to BAFF expression in renal podocytes

Figure 3. Probability of eGFR >90mL/min/1.73m2 at 60 months follow-up according to BAFF expression in renal interstitial inflammatory infiltrate

Figure 4. Probability of eGFR >90mL/min/1.73m2 at 60 months follow-up according to BCMA expression in renal interstitial inflammatory infiltrate

REFERENCES

1. Vivarelli M, Massella L, Ruggiero B, Emma F. Minimal change disease. Clin J Am Soc Nephrol. 2017;12: 332–345. doi:10.2215/CJN.05000516

2. Shabaka A, Ribera AT, Fernández-Juárez G. Focal Segmental Glomerulosclerosis: State-of-the-Art and Clinical Perspective. Nephron. Nephron; 2020. pp. 413–427. doi:10.1159/000508099

3. Maas RJ, Deegens JK, Smeets B, Moeller MJ, Wetzels JF. Minimal change disease and idiopathic FSGS: Manifestations of the same disease. Nature Reviews Nephrology. Nat Rev Nephrol; 2016. pp. 768–776. doi:10.1038/nrneph.2016.147

4. Noone DG, Iijima K, Parekh R. Idiopathic nephrotic syndrome in children. The Lancet. Lancet; 2018. pp. 61–74. doi:10.1016/S0140-6736(18)30536-1

5. Bajema IM, Wilhelmus S, Alpers CE, Bruijn JA, Colvin RB, Cook HT, et al. Revision of the International Society of Nephrology/Renal Pathology Society classification for lupus nephritis: clarification of definitions, and modified National Institutes of Health activity and chronicity indices. Kidney Int. 2018;93: 789–796. doi:10.1016/j.kint.2017.11.023

6. Food and Drug Administration. FDA approves first treatment for pediatric patients with lupus. Am Acad Pediatr News. 2019 [cited 2 Aug 2022]. doi:10.31525/fda2-ucm636756.htm

---

## [Decision Letter · Decision Letter 1]

4 Nov 2022

B-cell activating factor (BAFF) and its receptors’ expression in pediatric nephrotic syndrome is associated with worse prognosis

PONE-D-22-07630R1

Dear Dr. Tobón,

We’re pleased to inform you that your manuscript has been judged scientifically suitable for publication and will be formally accepted for publication once it meets all outstanding technical requirements.

Kind regards,

Frank JMF Dor, M.D., Ph.D., FEBS, FRCS

Academic Editor

PLOS ONE

Additional Editor Comments (optional):

Reviewers' comments:

Reviewer's Responses to Questions

**Comments to the Author**

1. If the authors have adequately addressed your comments raised in a previous round of review and you feel that this manuscript is now acceptable for publication, you may indicate that here to bypass the “Comments to the Author” section, enter your conflict of interest statement in the “Confidential to Editor” section, and submit your "Accept" recommendation.

Reviewer #2: All comments have been addressed

2. Is the manuscript technically sound, and do the data support the conclusions?

Reviewer #2: Yes

3. Has the statistical analysis been performed appropriately and rigorously? 

Reviewer #2: Yes

4. Have the authors made all data underlying the findings in their manuscript fully available?

Reviewer #2: Yes

5. Is the manuscript presented in an intelligible fashion and written in standard English?

Reviewer #2: Yes

6. Review Comments to the Author

Reviewer #2: The authors addressed to most of my concerns. I have no further comments. The main weakness is patient heterogeneity and mixed results. The value of the this manuscript is the elucidation of the relationship between BAFF and nephrotic syndrome in children.

7. PLOS authors have the option to publish the peer review history of their article (what does this mean?). If published, this will include your full peer review and any attached files.

Reviewer #2: No

---

## [Editor Report · Acceptance letter]

10 Nov 2022

PONE-D-22-07630R1 

B-cell activating factor (BAFF) and its receptors’ expression in pediatric nephrotic syndrome is associated with worse prognosis 

Dear Dr. Tobón:

I'm pleased to inform you that your manuscript has been deemed suitable for publication in PLOS ONE. Congratulations! Your manuscript is now with our production department. 

Kind regards, 

on behalf of

Dr. Frank JMF Dor 

Academic Editor

PLOS ONE